# Multiple Myeloma Associated Bone Disease

**DOI:** 10.3390/cancers12082113

**Published:** 2020-07-30

**Authors:** Stine Rasch, Thomas Lund, Jon Thor Asmussen, Anne Lerberg Nielsen, Rikke Faebo Larsen, Mikkel Østerheden Andersen, Niels Abildgaard

**Affiliations:** 1Department of Haematology, Odense University Hospital, Kloevervaenget 10, 12th Floor, DK-5000 Odense, Denmark; Stine.Rasch@rsyd.dk (S.R.); Thomas.Lund2@rsyd.dk (T.L.); Rikke.Faebo.Larsen@rsyd.dk (R.F.L.); 2Department of Internal Medicine, Division of Haematology, Sydvestjysk Sygehus, Finsensgade 35, DK-6700 Esbjerg, Denmark; 3Haematology Research Unit, Department of Clinical Research, University of Southern Denmark, Kloevervaenget 10, 12th Floor, DK-5000 Odense, Denmark; 4Department of Clinical Radiology, Odense University Hospital, Sdr. Boulevard 29, DK-5000 Odense, Denmark; jon.asmussen@rsyd.dk; 5Department of Nuclear Medicine, Odense University Hospital, Sdr. Boulevard 29, DK-5000 Odense, Denmark; anne.l.nielsen@rsyd.dk; 6Center for Spine Surgery & Research, Lillebaelt Hospital, Østre Hougvel 55, DK-5500 Middelfart, Denmark; Mikkel.Andersen2@rsyd.dk

**Keywords:** multiple myeloma, myeloma bone disease, pathophysiology, osteolysis, imaging, zoledronic acid, denosumab, vertebral augmentation, rehabilitation, exercise

## Abstract

The lytic bone disease is a hallmark of multiple myeloma, being present in about 80% of patients with newly diagnosed MM, and in more during the disease course. The myeloma associated bone disease (MBD) severely affects the morbidity and quality of life of the patients. MBD defines treatment demanding MM. In recent years, knowledge of the underlying pathophysiology has increased, and novel imaging technologies, medical and non-pharmaceutical treatments have improved. In this review, we highlight the major achievements in understanding, diagnosing and treating MBD. For diagnosing MBD, low-dose whole-body CT is now recommended over conventional skeletal survey, but also more advanced functional imaging modalities, such as diffusion-weighted MRI and PET/CT are increasingly important in the assessment and monitoring of MBD. Bisphosphonates have, for many years, played a key role in management of MBD, but denosumab is now an alternative to bisphosphonates, especially in patients with renal impairment. Radiotherapy is used for uncontrolled pain, for impeding fractures and in treatment of impeding or symptomatic spinal cord compression. Cement augmentation has been shown to reduce pain from vertebral compression fractures. Cautious exercise programs are safe and feasible and may have the potential to improve the status of patients with MM.

## 1. Introduction

Multiple myeloma is an incurable B-cell malignancy characterized by proliferation and expansion of clonal plasma cells in the bone marrow [1]. The presence of osteolytic lesions is a hallmark of multiple myeloma and occurs in up to 80% of patients at diagnosis [2]. The axial skeleton, particularly the spine, and the proximal long bones, are most often affected, but any bone can be involved [3]. Myeloma bone disease also includes hypercalcemia, pathological fractures, bone pain and risk of spinal cord compression, all of which are associated with reduced quality of life [4,5]. Furthermore, skeletal-related events may have a negative impact on survival [6,7]. Despite the new, more targeted anti-myeloma treatments, which have significantly improved the overall survival for patients with multiple myeloma [8,9], MBD remains a major problem [10].

## 2. Pathophysiology

Bone remodeling is a continuous, lifelong process where old bone is resorbed by osteoclasts and replaced by new bone created by the osteoblasts. The process is well balanced and mediated by crosstalk between osteoblasts, osteoclasts, osteocytes, immune cells and bone matrix bound factors, and is partly mediated by certain cytokines and hormones [11]. In patients with MBD, the harmonious coupling of osteoclast and osteoblast activity is lost. Increased osteoclast activity and suppressed osteoblast activity lead to increased bone resorption that is not compensated for by bone formation [12].

A crucial regulatory system of bone remodeling is the receptor activator of nuclear factor kappa B (RANK)/RANK ligand (RANKL) signaling pathway. Through the RANK receptor on the precursor osteoclasts, RANKL stimulates recruitment, differentiation and activity of the osteoclasts. The bone marrow stromal cells (BMSC) and osteoblasts secrete osteoprotegerin (OPG), a decoy receptor for RANKL, which inactivates RANKL, thereby reducing osteoclast activation [13,14]. Myeloma cells interact with the bone marrow microenvironment, activating molecular cascades that lead to increased RANKL and decreased OPG expression [15,16]. Consequently, RANKL/OPG ratio is increased as the key element in the increased osteoclast hyper-activation.

Secondly, osteoblast inhibition, and thereby reduced bone formation, plays an important role in the severity of MBD. Several factors are involved in downregulation of osteoblastic activity by interfering with the Wingless (Wnt)/(DKK1) signaling pathway, which is a key pathway for osteoblast recruitment and activation [17]. Dickkopf-1 (DKK1), expressed by the myeloma cells and BMSC, antagonizes the WNT-pathway, blocks the differentiation of osteoblasts, and high DKK1 expression in the bone marrow is associated with more severe MBD [18,19,20].

Besides the signaling abnormalities involved in the control of osteoclast and osteoblast activity, it has been suggested that direct myeloma cell invasion into the bone remodeling compartment is involved in the pathophysiology [21]. The remodeling compartment is a closed microenvironment, which is shielded against the bone marrow space by a thin canopy. It has been shown that these canopies may be infiltrated and disrupted by myeloma cells, thereby causing uncoupling of the normal remodeling process [21].

Figure 1 summarizes the key pathophysiological abnormalities in MBD. Beside the abovementioned pathways, many other molecular pathways and signaling molecules are hypothesized to be involved in the pathophysiology of MBD, and some data even indicate that the involved mechanisms may differ between patients as summarized in a thorough, recent review [22]. Understanding these mechanisms is crucial to improve the management of MBD.

## 3. Imaging

Imaging plays a crucial role when diagnosing multiple myeloma (MM). First of all, identification of lytic lesions is one of the CRAB-criteria (Calcium, Renal, Anemia, Bone) that define organ damage and the need for starting anti-myeloma therapy [23]. Imaging is also essential to distinguish solitary plasmacytoma from multiple myeloma, and for identifying extramedullary disease [24,25]. Finally, imaging is increasingly important in post-treatment response evaluation [26].

### 3.1. Definition of Myeloma Associated Bone Disease

In 2014, the International Myeloma Working Group (IMWG) updated the criteria for the diagnosis of multiple myeloma and stated that one or more typical punched-out lytic bone destructions (≥5 mm in size) on CT/low-dose CT or PET/CT would meet the CRAB-criteria regardless of its visualization on skeletal radiography [27]. Increased focal FDG uptake on PET-CT alone is not sufficient to define bone disease; evidence of lytic bone destruction must be present on the CT-part. The presence of osteoporosis or vertebral compression fractures in the absence of lytic lesions is not evidence of MBD. Additionally, more than one focal lesion on magnetic resonance imaging (MRI), reflecting “tumoral” changes in the bone marrow, fulfils the imaging criteria for treatment-demanding MM [27]. Both MRI and PET/CT are able to detect what is referred to as focal lesions, however only lytic bone lesions detected by CT are truly evidence of MBD [28].

### 3.2. From Conventional Skeletal Survey to Whole-Body CT

Conventional skeletal survey (CSS) has been the standard imaging technique in the radiological diagnosis of multiple myeloma for many years [29]. A definite advantage of CSS has been its general availability and low cost. However, CSS has limitations, especially in relation to sensitivity. An older study from 1967 [30] showed that lytic bone disease only becomes detectible by CSS when over 30% of the trabecular bone is lost.

Particularly in the spine and pelvis, whole-body low dose CT (WBLDCT) has been shown to have superior sensitivity in detecting osteolytic lesions. For instance, superimposed air in the bowel can challenge the interpretation of the pelvis (Figure 2). In a study of 32 patients with MM, it was shown that osteolytic lesions in the pelvis or spine were found in 50% of the patients examined with radiographs, and in 74% of patients examined with WBLDCT [31]. A large, retrospective, international, multicenter study performed a blinded comparison of CSS and WBLDCT in patients with newly diagnosed MM [32]. In general, WBLDCT was superior to CSS in identifying lytic lesions. However, the difference in the sensitivity depended on the location of the lytic lesions. WBLDCT was superior in detecting lesions in the spine and pelvis, whereas no significant difference in sensitivity was observed in long bones. In a large sub-cohort of patients with apparent smoldering MM (SMM), lytic lesions were identified by WBLDCT, but not by CSS, in 22.2% of the patients. These patients had a higher probability of progression to symptomatic myeloma compared to those without bone destructions [32]. These and similar, small cohort study observations caused a change in diagnostic practice in many MM centers. WBLDCT was implemented as the standard for diagnostic screening for MBD. Also, in the updated IMWG 2014 guideline, WBLDCT was recommended over CSS [27].

The appendicular bone marrow consists partly of adipose tissue, but in multiple myeloma patients, the bone marrow is diffusely or focally infiltrated by neoplastic plasma cells to varying degrees. Bone marrow changes are traditionally mostly investigated and reported by magnetic resonance imaging techniques (see below), but nodular or diffuse infiltration of long bones can also be detected by WBLDCT and has been reported to have prognostic significance. Identified focal and diffuse pattern in the appendicular bone marrow by WCLDCT is associated with a shorter PFS and OS [33].

Today, WBLDCT is considered standard of care in diagnostic screening for MBD [28]. If WBLDCT is not available, CSS can still be used [28].

### 3.3. MRI as a Diagnostic and Prognostic Tool in Patients with Multiple Myeloma

Magnetic resonance imaging (MRI) has the ability to detect early focal and diffuse infiltration patterns of the bone marrow [34]. Studies have shown that MRI, either axial or whole body, has a higher sensitivity in detecting bone marrow involvement in multiple myeloma compared to CSS and WBLDCT [35,36,37]. Thus, a study of 611 patients concluded that MRI was able to detect more focal lesions than CSS, and the presence of more than seven focal lesions on MRI was an independent adverse feature for survival [36]. However, it should be noticed that a focal lesion in the bone marrow on MRI is not evidence of an established lytic destruction; it reflects a dense cellular infiltration that may or may not have a connected lytic lesion, or may (or may not) precede development of a lytic lesion. Lytic destruction is identified by loss of bone on CT or radiographs. Thus, it is important to realize that MRI and CT offer complementary information in many patients [38].

In line with this, MRI may identify focal lesions in patients with presumed SMM and normal WBLDCT. Two independent studies found that the finding of more than one focal lesion on axial or whole-body MRI was associated with a 70–80% risk of progression to symptomatic disease within 2 years [39,40]. Based on this observation, the IMWG included the criteria “more than one focal lesion on MRI” in the updated 2014 criteria for treatment demanding disease [27]. Therefore, whole-body MRI should be the next diagnostic procedure in a patient with normal findings on WBLDCT and no other CRAB-criteria. This patient would traditionally have been diagnosed as a SMM patient; however, whole body MRI may up-classify the patient to have treatment-demanding disease. However, it should be realized that “more than one focal lesion” on MRI is not an unequivocal finding; MRI findings are not specific, and there will be a role for interpretation. Dubious findings may require confirmation by biopsy, or a wait-and-watch strategy with repeated MRI after 3-6 months. Progression of focal lesions or appearance of new focal lesions identify a subgroup of patients with true active disease, whereas unchanged findings indicate low risk and SMM phenotype [41]. In contrary to focal lesions, diffuse infiltration of the bone marrow on MRI is not considered a myeloma-defining event, but should lead to follow-up imaging in 3–6 month [27].

Figure 3 illustrates typical findings on whole-body MRI (WBMRI). WBMRI is recommended over combined spinal and pelvic MRI as lesions in rib cage, shoulder girdles and long bones could otherwise be missed.

The NICE-guidelines suggest considering whole-body MRI as first-line imaging when multiple myeloma is suspected [42]. At least in particular clinical settings MRI will be the preferred methodology. Whole-body MRI is recommended as the first choice in patients with suspected solitary bone plasmacytoma (whereas FDG-PET/CT is recommended in suspected solitary extramedullary plasmacytoma) [28] and MRI is recommended as the first-line investigation if spinal cord compression is suspected and is the chosen imaging technique to characterize whether vertebral compression fractures are caused by osteopenia only or are myeloma infiltrated [43,44].

### 3.4. The Evolving Role of FDG-PET/CT in Multiple Myeloma

Positron Emission Tomography (PET)/CT using ^18^Fluoro-deoxy-glucose (FDG) as the radioactively labelled tracer (FDG-PET/CT) permits whole-body assessment and is able to visualize both extramedullary and skeletal disease. FDG-PET offers dynamic information on metabolic active sites of disease, and CT contributes with precise anatomic information, thereby making the combined investigation able to identify and differentiate between active and inactive sites and provide information about extramedullary involvement [45]. Due to the CT part, PET/CT is superior to CSS in diagnosing lytic bone lesions [46]. Compared to MRI, PET/CT has a lower sensitivity for detection of bone marrow involvement [46]. A recent systematic review compared whole-body MRI and FDG PET/CT in their ability to detect myeloma skeletal lesions and suggested that MRI is more sensitive but less specific than FDG PET/CT. Yet, it also concluded that most of the included studies were heterogeneous and lacking an independent reference standard [47].

However, several studies have shown that PET-positive lesions offer prognostic information, both at diagnosis, during and after treatment. The number of lesions, the intensity of tracer uptake, and the presence of extra-medullary disease has been shown to be associated with inferior survival [48,49,50,51]. In the response criteria of minimal residual disease negativity, FDG PET/CT is included and requires disappearance of abnormal tracer uptake found on baseline scan or decrease to less than mediastinal blood pool or surrounding normal tissue [26].

The IMWG recommends that PET/CT can be used in place of WBCT, but also in place of WBMRI if imaging with MRI is not possible [28].

Sodium ^18^F-Fluoride (NaF) is a bone-seeking agent introduced in 1962 [52]. The uptake of ^18^F-fluoride reflects blood flow and osteoblastic activity and thereby bone remodeling [53,54]. NaF-PET is used in the assessment of malignant and benign skeletal disease and has been suggested as a potentially valuable tool in the assessment of MM as well [53,55]. Hypothetically, post-treatment NaF-PET could identify bone healing activity in lytic lesions [25]. However, so far, studies have not been able to demonstrate that NaF-PET provides additional clinical information when assessing MBD or evaluating treatment response compared to FDG-PET [56,57,58]. Figure 4 shows typical findings on FDG-PET/CT and NaF-PET/CT in the same patient and illustrates how the findings differentiate.

Other PET tracers, such as choline-based tracers, have been proposed for PET/CT imaging in patients with MM. ^11^C-Choline and ^18^F-Fluorocholine PET/CT were initially developed for prostate cancer imaging [59]. Choline is actively incorporated into the new cell membranes [60]. Results from two smaller studies suggest that Choline PET/CT detects up to 75% more focal lesions than FDG PET/CT in patients with MM suspected of progression or relapse [61,62]. Thus, potentially there is a value in using other tracers than FDG in MM; however, this needs to be explored further and validated in clinical trials.

### 3.5. Follow-Up, Response Assessment, and Relapse

At the moment, there are no clear recommendations regarding routine follow-up, but in general, CSS should not be used for disease monitoring [42]. It is recommended to repeat relevant imaging of the same modality, PET/CT or WBMRI, as part of response evaluation in patients where active disease sites or extramedullary disease were identified prior to start of therapy. In patients with known extramedullary manifestations, imaging must be repeated for response assessment. Oppositely, for now, there is no consensus that whole body imaging should be performed as part of response evaluation in all patients. However, in patients with achieved complete remission after high-dose chemotherapy and autologous stem-cell transplantation (ASCT), PET-positivity may persist and predict early relapse and inferior outcome [63]. Moreover, IMWG has included FDG-PET/CT into response assessment when evaluating MRD status [26] and recommends PET/CT assessment at baseline and for response assessment in all patients included in clinical trials [28]. If PET/CT is not available, diffusion-weighted WBMRI can be used and has shown some promising ability to assess response to therapy [64].

For response assessment with FDG PET/CT as well as with WBMRI it applies that there is a continued need for standardization of the techniques, clear definition of response criteria and prospective evaluation hereof.

## 4. Medical Treatment

### 4.1. Bisphosphonates

Since Berenson’s pamidronate trial in 1996, bisphosphonates have played a key role and been standard of care in management of MBD [65]. Bisphosphonates are pyrophosphate analogues that bind to bone and are ingested by the osteoclasts, leading to inhibition of osteoclastic activity. There are different types of bisphosphonates: Pamidronate, alendronate, ibandronate and zoledronate are all examples of nitrogen-containing bisphosphonates and inhibit the mevalonate pathway. Non-nitrogen-containing bisphosphonate, like clodronate, results in accumulation of hydrolytical stable analogues of adenosine triphosphate. Zoledronate, pamidronate and clodronate have been most intensively studied in MM. Both types of bisphosphonates cause inhibition and apoptosis of osteoclasts. Furthermore, data indicate that bisphosphonates, in addition to their bone-protective effects, may have antitumor activity due to an uncoupling of the hypothesized vicious circle between bone resorption and tumor growth in MM [66].

Few prospective, randomized trials comparing the different bisphosphonates head to head have been conducted. The Rosen study from 2003 [67] compared zoledronic acid to pamidronate, in patients with either MM or breast cancer. Zoledronic acid was superior to pamidronate in reducing the risk of skeletal related events (SRE), but the subgroup analysis only found a significant difference in the breast cancer population. No data on overall survival (OS) were provided. The UK MRC Myeloma XI study from 2011 compared zoledronic acid with clodronate [68,69]. Zoledronic acid was found to be superior to clodronate both in regard to SRE and overall survival. The lower risk of SRE was also observed in patients without bone lesions at baseline [69].

A meta-analysis by the Cochrane database from 2017, including 24 randomized controlled trials with a total of 7293 patients, investigated the beneficial and adverse effects associated with the use of different types of bisphosphonates in patients with MM [70]. They concluded that bisphosphonates reduce overall fractures and pain, and that zoledronic acid improves overall survival compared to no bisphosphonate treatment. The meta-analysis showed no significant difference between the different types of bisphosphonate [70]. In contrast, a retrospective cohort study, of over 1000 patients who had been treated with either zoledronic acid or pamidronate, reported that zoledronic acid compared to pamidronate reduced the risk of SRE by 25% and was associated with an increased overall survival [71]. The current recommendation by IMWG is to initiate treatment with either zoledronic acid or pamidronate in all patients with symptomatic MM, regardless of detectible osteolytic lesions on baseline imaging [72].

In patients with smoldering myeloma, it has not been shown that bisphosphonates prolong the time to progression to symptomatic disease [73,74] and it is therefore not recommended.

The optimal duration of bisphosphonate treatment is still controversial. In most randomized, controlled trials, bisphosphonates were administered up to 2 years. In the Myeloma IX trial however, bisphosphonates were given until progression. A sub-analysis conducted in patients receiving treatment from year 2 and onward demonstrated persistent superiority of the more potent zoledronic acid, both in regard to SRE and OS [75]. Interestingly, the cumulative incidence of renal complication and osteonecrosis of the jaw (ONJ) seemed to reach a plateau between year 2 to 3 [76]. Another group investigated if 4 years treatment with zoledronic acid was superior to treatment for only 2 years. Prolonged treatment reduced SRE but no difference in OS was observed [77]. Some experts argue for less bisphosphonate treatment in cases where the myeloma is well treated [78]. Indeed, data from the Myeloma IX trial showed that a reduction in SRE was not observed in patients achieving at least CR after ASCT, and that no survival benefit was seen in patients achieving VGPR or better after ASCT [79].

All the referred studies used the standard dosing of pamidronate and zoledronic acid every 3–4 weeks. However, an open-label study by Himelstein et al. comprised 1154 patients with bone metastases, including 278 patients with MM, and compared zoledronic acid administrated every 4 weeks to every 12 weeks for up to 2 years [80]. No differences in SRE or side effects were observed. Unfortunately, the study had a relatively high drop-out rate of 31%, and because only about 25% of the included patients had MM, it is difficult to draw firm conclusions about the possible adjustment of zoledronic acid scheduling in MM. Other groups have proposed that the interval between zoledronic acid infusions could be guided by the levels of the bone resorption marker Ntx-1 (N-terminal telopeptide of type 1 collagen) in the urine. [81]. Though this strategy is appealing and could reduce the risk of developing ONJ, the evidence for doing this is still insufficient.

IMWG recommends that in patients who do not achieve very good partial response or better, zoledronic acid should be administrated monthly until disease progression [72]. Otherwise, it is suggested that bisphosphonates should be administered for up to 2 years and should be reinitiated at relapse, if discontinued earlier [72]. Rationally, and because bisphosphonate treatment is prophylactic, re-initiation of zoledronic acid should be at biochemical relapse and not postponed until clinical relapse. This is supported by a Spanish study that randomized patients to zoledronic acid versus no bisphosphonate at first sign of biochemical relapse. Although no effects were demonstrated on time to need of treatment or survival, the patients who were re-initiated early with zoledronic acid had less SREs at the time of treatment demanding relapse [82].

As mentioned, a serious but rare adverse event of bisphosphonate use is ONJ. Recent, randomized, controlled trials showed an incidence of 3–4% in myeloma patients receiving zoledronic acid [68,83]. The median time from start of treatment to ONJ was found to be 13.6 months [83]. Invasive dental procedures, dental prostheses and intravenous bisphosphonate administration and long-term treatment as well as the myeloma itself are all risk factors associated with ONJ [84]. A case-control study showed that patients, who were assessed by their dentist and had all necessary dental procedures done before initiating treatment with zoledronic acid, had a three-fold decrease in the risk of developing ONJ [85]. If invasive dental procedures are required during bisphosphonate treatment a “drug holiday” before and after invasive dental procedures is generally recommended [72], despite the fact that bisphosphonates remain in the skeleton for many years [86,87]. A retrospective study indicated that prophylactic antibiotics during invasive dental procedures may reduce the risk of developing ONJ [88]. Bisphosphonate-induced nephrotoxicity is another major concern when treating patients with MM. Zoledronic acid should be dose reduced already with a mild to moderate renal impairment (CrCl 30–60 mL/min) and is not recommended in patients with severe renal impairment (<30 mL/min). A recent publication, including patients from 5 European countries, showed that 51 % of all patients had renal insufficiency at the start of first line treatment, and 3% had severe renal impairment [89]. The study also found that a quarter of the patients with sufficient renal function never started bisphosphonate treatment.

### 4.2. Denosumab

For patients with renal impairment and normal renal function, denosumab could be a viable alternative to bisphosphonates. Denosumab is a human monoclonal antibody that binds to and inhibits RANKL signaling and thereby blocks osteoclast activation [90]. It is not excreted through the kidneys, but degraded by endocytosis. Results from a large, randomized, controlled, phase 3 study, including 1718 patients with MM, showed that denosumab was non-inferior to zoledronic acid in the time to first SRE and OS. Analysis of the exploratory progression-free survival (PFS) endpoint favored denosumab, and this somewhat puzzling observation has been further analyzed [91]. The PFS benefit was restricted to the patients planned for (undergoing?) ASCT. One hypothesis could be that RANKL signaling is involved in re-activation of “dormant” myeloma cells [92]. The observation that the PFS improvement was restricted to younger ASCT-eligible patients indicates that it could be the myelo- and stroma-ablative high-dose Melphalan in combination with denosumab that is beneficial.

The incidence of ONJ was the same for denosumab and zoledronic acid, but a higher incidence of hypocalcemia was observed among patients treated with denosumab. Denosumab was given every 4 weeks, like zoledronic acid. The study only included patients without renal impairment, and all patients had osteolysis at diagnosis. Sparse data exist on the safety of denosumab in patients with MM with renal insufficiency. In patients with bone metastases and severe renal insufficiency, denosumab can be given, but causes an increased risk of electrolyte deficiencies [93]. Unlike bisphosphonates, denosumab is not incorporated in the bone matrix and its effect declines rapidly after cessation [94]. This could be a benefit in regard to “drug holidays” prior to invasive dental procedures. Indeed, murine data indicate that ONJ may heal better after cessation of denosumab compared to zoledronic acid [95]. The downside of this rapid cessation of effect is that a rebound effect has been observed in patients with osteoporosis, where increased bone loss has been observed when treatment is discontinued, presumably because of compensatory upregulated RANKL signaling during denosumab treatment [96]. At the moment, there are no data on how to stop treatment with denosumab in patients with MM, but it has been suggested to switch treatment to bisphosphonate or to end the treatment with a single dosing of zoledronic acid [97].

## 5. Non-Pharmaceutical Treatment

### 5.1. Radiotherapy

Historically, radiotherapy has played an important part in managing MBD. One of the most common indications for radiotherapy is pain reduction. A retrospective study found that up to 84% of patients with myeloma obtained pain relief after radiotherapy [98]. Other indications are prophylactic treatment of impending pathological fractures, spinal cord compression and management of local neurological symptoms [99]. For patients with myeloma with spinal cord compression, radiotherapy alone offers excellent response rates (97%), local control (93% at 1 year, 82% at 2 years) and functional outcomes (64% of non-ambulatory regained the ability to walk) [100].

Some concerns regarding depletion of the bone marrow reserve after concurrent chemotherapy and radiotherapy exist. A smaller study, including 39 patients with myeloma receiving radiotherapy alone or radiotherapy with concurrent novel agents-based chemotherapy, concluded that concurrent treatment with radiotherapy and systemic treatment was safe regarding hematologic toxicity and was well tolerated in the majority of patients (87.5%) [101].

### 5.2. Vertebral Augmentation

Vertebroplasty and kyphoplasty are both minimal invasive fluoroscopic guided percutaneous surgical procedures used to reduce pain caused by vertebral compression fractures in patients with myeloma. Cement augmentation of the spine is possible at all spinal levels. In the cervical region, the vertebral bodies can be accessed through an anterior approach. Thoracic and lumbar vertebrae are reached through a transpedicular approach with a Jamshidi needle. Bone cement (polymethylmethacrylate) is injected into the vertebral body under imaging guidance. Kyphoplasty differs from vertebroplasty as the height of the fractured vertebra is restored with an inflatable balloon catheter prior to injection of bone cement. The void created by the balloon catheter while restoring the vertebral height allows for more controlled delivery of cement, reducing the risk of bone cement leakage.

Both procedures can be performed under local anesthesia in an outpatient setting. However, kyphoplasty is often performed under general anesthesia as some patients experience pain while the vertebral height is restored. For patient safety reasons, the procedure is performed under local anesthesia, allowing the patient to communicate radiating pain, which could indicate that the needles are out of target, thereby minimizing the risk of neurological injury. Figure 5 illustrates typical lumbar spine MRI findings prior to vertebroplasty, and the final radiological appearance after the procedure.

A randomized, controlled trial, including 134 cancer patients with vertebral fractures, of whom 49 had multiple myeloma, found that kyphoplasty resulted in significant pain relief, improved back-specific functional status, quality of life (QoL) and self-reported physical activity, compared to non-surgical management. These improvements persisted throughout the entire study period until the end of the study at 12 months [102]. Similar results with reduced pain and improved QoL after cement augmentation by vertebroplasty or kyphoplasty have been reported in MM cohort studies [103,104,105,106,107].

A meta-analysis from 2014 found vertebroplasty and kyphoplasty to be equally effective in reducing pain in myeloma patients [104]. In favor of kyphoplasty is a potentially better correction of the patients’ sagittal balance. Bone cement leakage is commonly reported (4–26% per treated vertebra), but is mostly asymptomatic [108].

A retrospective analysis, with 18 myeloma patients who underwent vertebroplasty prior to autologous stem cell transplant showed that vertebroplasty could be done without affecting peripheral blood stem cell collection and transplant [109]. The current recommendation in myeloma associated vertebral collapse is to consider vertebral augmentation if it causes moderate or severe pain, and particularly if it affects mobilization [110] This is supported by a recently published national guideline based on the GRADE-approach [111] recommending vertebral augmentation as treatment in patients with painful vertebral lesions and malignant hematologic disease [112].

### 5.3. Rehabilitation and Exercise

Exercise has been demonstrated to have a significant beneficial effect on QoL and physical function in patients with cancer [113,114], but only few studies have been conducted on patients with MM [115,116]. This is probably explained by the MBD and suspected increased risk of pathological fractures. However, two literature reviews found that exercise appeared safe and acceptable for patients with MM, but also concluded that data are limited and that no conclusion regarding the effectiveness of exercise could be drawn [115,116]. All studies in patients with MM have been conducted in patients before, during or after ASCT.

Baseline data from a randomized controlled trial indicate that patients with newly diagnosed multiple myeloma generally had lower physical function compared to the normal population, and this goes particularly for patients with bone disease and fractures [117]. A feasibility study, evaluated 30 patients with newly diagnosed myeloma who were randomized 1:1 to usual care or usual care and individualized, supervised exercise combined with home-based exercise for 10 weeks. Sixty-seven percent of the patients had bone involvement. The study showed that even in older patients and in patients with MBD, individualized physical exercise is feasible and safe around the time of diagnosis [118]. The following expanded effect trial included 100 patients with newly diagnosed MM in a randomized setting did not show effect on physical function, physical activity, QoL, or pain [119]. However, the results of physical function indicated a trend for less loss of muscle strengths in the intervention group, but there is a need to pay attention to pain, since this might be worsened by the intervention [119].

## 6. Conclusions

Despite improved anti-myeloma treatments, MBD remains a significant problem. The understanding of the pathophysiology has improved and may lead the way for development of new bone directed treatments. Until then, anti-resorptive treatment with bisphosphonates or denosumab is standard of care. Modern imaging with CT, PET/CT, and MRI play an essential role in diagnosing and monitoring MBD and help to guide supplementary treatment with irradiation and vertebral augmentation. Exercise in patients with MM is safe and feasible when relevant restrictions are taken into account; however, so far, no studies have demonstrated definite benefit of training.

## Figures and Tables

**Figure 1 cancers-12-02113-f001:**
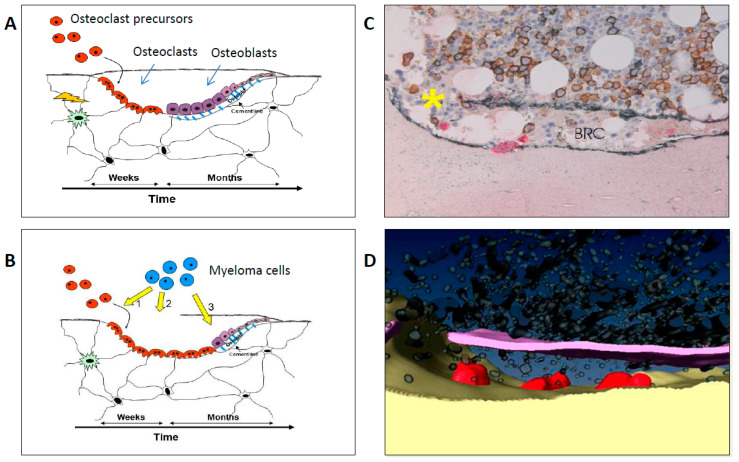
(**A**) Cartoon illustrating the normal bone remodeling taking place in bone remodeling compartments (BRC) that are separated from the bone marrow environment by a thin roofing canopy. (**B**) summarizes the major pathophysiological events in myeloma bone disease: (1) The multiple myeloma (MM) cells increase recruitment of osteoclast precursors, (2) MM cells infiltrate the BRC, disrupt the canopy and stimulate osteoclast activity, and (3) MM cells inhibit the osteoblasts, cause osteoblastopenia, and MM cell invasion into the BRC contributes to the uncoupling of osteoclast and osteoblast activity, (**C**) shows the microscopic findings where MM cells (brown) disrupt the canopy (yellow asterisk) and invade into the BRC. (**D**) A computerized reconstruction of canopy disruption and invasion of MM cells into the BRC. (**C**,**D**) are reproduced by permission from the original work published in British Journal of Haematology from 2010 [21].

**Figure 2 cancers-12-02113-f002:**
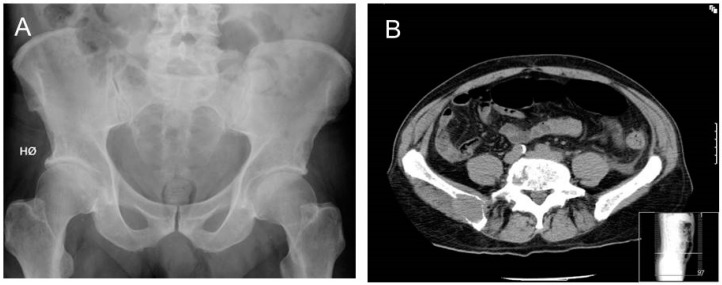
(**A**) A radiograph of the pelvis is assessed by the radiologist as normal. (**B**) CT of the pelvis in the same patient identifies a large lytic lesion with soft tumor in right crista region. Super-imposed air in the bowel hides the destruction on the conventional radiograph.

**Figure 3 cancers-12-02113-f003:**
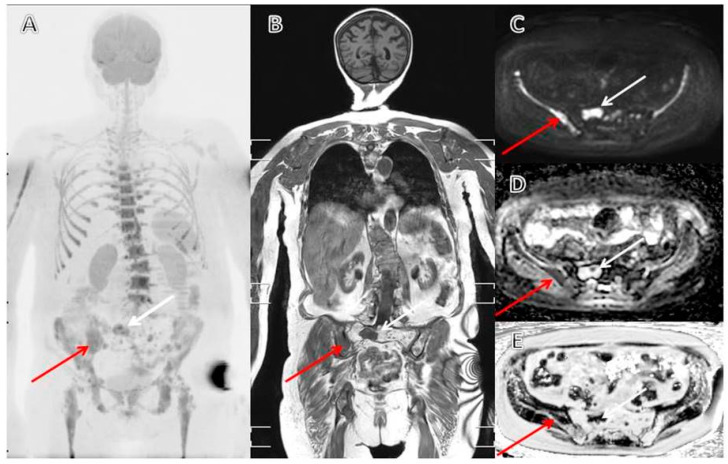
Whole body MRI of relapsing Myeloma, multiple new lesions primarily in right arm, spine, ribs and right side of pelvis. Many previously treated lesions in spine, pelvis and legs. Red arrow: typical new lesion with myeloma cells (low, homogenous ADC), White arrow: typical old lesion with cell free water content (high ADC) and possible focal recurrence. (**A**) MIP of DWI-sequence with high b-value, (**B**) T1-DIXON in-phase sequence, (**C**) DWI-sequence high b-value, (**D**) ADC (parametric map calculated from DWI), and (**E**) Fat fraction (parametric map calculated from T1-DIXON).

**Figure 4 cancers-12-02113-f004:**
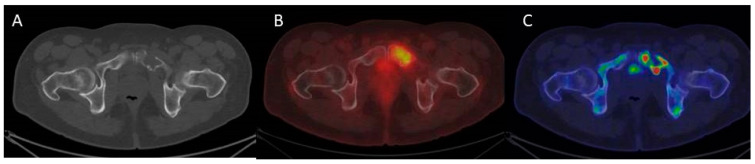
(**A**) CT of the pelvis showed a lytic lesion of the left pubic area. (**B**) ^18^F-FDG PET/CT demonstrated increased metabolism localized within the osteolytic lesion consistent with myeloma cells. (**C**) PET/CT with ^18^F-Sodium Fluoride (NaF) revealed increased patchy uptake in the periphery of the lytic lesion indicating areas bone remodeling activity.

**Figure 5 cancers-12-02113-f005:**
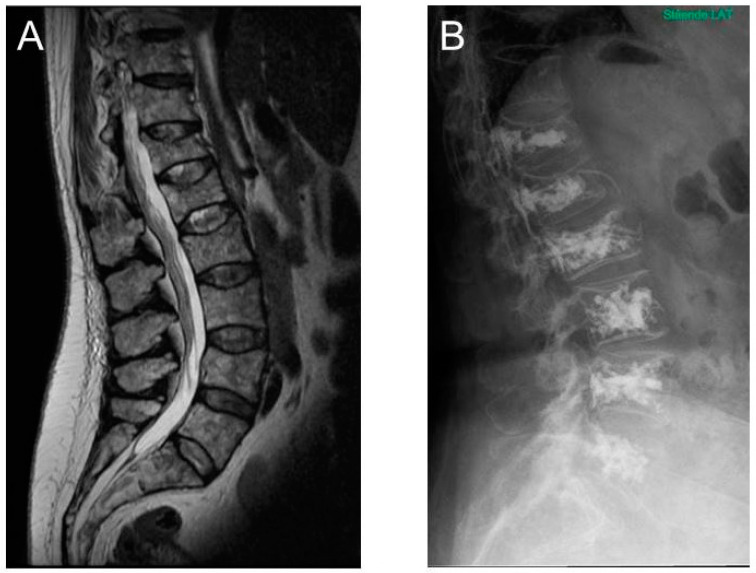
(**A**) MRI scan of multiple myeloma patient with several lesions of lumbar vertebrae and Th12. (**B**) Post-operative X-ray of the same patient after vertebroplasty in Th12 to L5 vertebrae. Bone cement leakage is visible related to L1.

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
