# Peer review of "Multiple Myeloma Associated Bone Disease"

_cancers, 2020, doi:10.3390/cancers12082113_

Round 1

Reviewer 1 Report

Rasch et al have described the current knowledge of myeloma bone disease in their manuscript entitled " Multiple Myeloma associated Bone Disease". The review manuscript is well organized, written in standard English and well summarizes the recent advances in myeloma bone disease research and treatment. One minor issue should be improved for publication.

(Minor comment)

1. In figure 1A, B, the authors should describe what each cell stands for.

Author Response

We have revised Figure 1A and B according to the reviewers suggestion.

Reviewer 2 Report

This is a through and well written review of multiple myeloma associated bone disease. Minor suggestions are listed below:

  1. I think that it looks like some mistake to describe the sentence. Please check the following sentence on last part of pathophysiology in Page 2: “some data even indicate that the involved mechanisms may differ between patients Please find more information in a through, recent review”
  1. On page 5, “Whole body MRI is recommended as the first choice in patients with suspected solitary plasmacytoma”. In fact, IMWG guideline recommends ‘FDG PET/CT for patients with extramedullary solitary plasmacytoma’. It would be better to describe it more accurately.  

Author Response

1) I think that it looks like some mistake to describe the sentence. Please check the following sentence on last part of pathophysiology in Page 2: “some data even indicate that the involved mechanisms may differ between patients Please find more information in a through, recent review”

Answer: We have changed the sentence on page 2 to: “..some data even indicate that the involved mechanisms may differ between patients as summarized in a thorough recent review”.

2) On page 5, “Whole body MRI is recommended as the first choice in patients with suspected solitary plasmacytoma”. In fact, IMWG guideline recommends ‘FDG PET/CT for patients with extramedullary solitary plasmacytoma’. It would be better to describe it more accurately.

Answer: In IMWG 2019 imaging guideline “International myeloma working group consensus recommendations on imaging in monoclonal plasma cell disorders” (Lancet Oncology 2019 Jun; 20(6): e302-312 (ref 28)) the following is stated concerning recommended imaging in solitary plasmacytoma: “We recommend MRI of the spine and pelvis (or whole body MRI, if available) in patients with newly diagnosed solitary bone plasmacytoma. We recommend whole body FDG PET-CT in patients with newly diagnosed solitary extramedullary plasmacytoma. If MRI is not available, WBCT or FDG PET-CT can be used an alternative in patients with newly diagnosed solitary bone plasmacytoma”.We have clarified this in text on page 6: “Whole-body MRI is recommended as the first choice in patients with suspected solitary bone plasmacytoma (whereas FDG-PET/CT is recommended in suspected solitary extramedullary plasmacytoma) [28]

Thus, MRI is recommended for solitary bone plasmacytoma whereas FDG-PET/CT is recommended for solitary extramedullary (non-bone) plasmacytoma.

We have clarified this in text on page 6: “Whole-body MRI is recommended as the first choice in patients with suspected solitary bone plasmacytoma (whereas FDG-PET/CT is recommended in suspected solitary extramedullary plasmacytoma) [28]